# PIKfyve, expressed by CD11c-positive cells, controls tumor immunity

Jae Eun Choi[1,2,11], Yuanyuan Qiao[1,2,3], Ilona Kryczek[4,5], Jiali Yu[4,5], Jonathan Gurkan[1,2,12], Yi Bao[1,2], Mahnoor Gondal[1,2,6], Jean Ching-Yi Tien[1,2], Tomasz Maj[4,5], Sahr Yazdani[1,2,13], Abhijit Parolia[1,2], Houjun Xia[4,5], JiaJia Zhou[4,5], Shuang Wei[4,5], Sara Grove[4,5], Linda Vatan[4,5], Heng Lin[4,5], Gaopeng Li[4,5], Yang Zheng[1,2], Yuping Zhang[1,2], Xuhong Cao[1,2,7], Fengyun Su[1,2], Rui Wang[1,2], Tongchen He[1,2], Marcin Cieslik[1,2,6], Michael D. Green[5,8,9], Weiping Zou[1,2,3,4,5] ✉ & Arul M. Chinnaiyan[1,2,3,7,10] ✉

Cancer treatment continues to shift from utilizing traditional therapies to targeted ones, such as protein kinase inhibitors and immunotherapy. Mobilizing dendritic cells (DC) and other myeloid cells with antigen presenting and cancer cell killing capacities is an attractive but not fully exploited approach. Here, we show that *PIKFYVE* is a shared gene target of clinically relevant protein kinase inhibitors and high expression of this gene in DCs is associated with poor patient response to immune checkpoint blockade (ICB) therapy. Genetic and pharmacological studies demonstrate that PIKfyve ablation enhances the function of CD11c⁺ cells (predominantly dendritic cells) via selectively altering the non-canonical NF-κB pathway. Both loss of *Pikfyve* in CD11c⁺ cells and treatment with apilimod, a potent and specific PIKfyve inhibitor, restrained tumor growth, enhanced DC-dependent T cell immunity, and potentiated ICB efficacy in tumor-bearing mouse models. Furthermore, the combination of a vaccine adjuvant and apilimod reduced tumor progression in vivo. Thus, PIKfyve negatively regulates the function of CD11c⁺ cells, and PIKfyve inhibition has promise for cancer immunotherapy and vaccine treatment strategies.

The success of immunotherapy has fundamentally altered our understanding of cancer and changed the standard of care for cancer treatment[1,2]. Understanding the mechanisms of immunotherapy efficacy and resistance are needed to further improve cancer outcomes[3,4]. Attention has centered on the countless ways in which tumor-associated antigen (TAA)-specific T cells are directly or indirectly suppressed[4–8]. However, fewer studies have investigated the contribution of professional antigen presenting cells (APCs), such as dendritic cells (DC), to antitumor immunity. DCs prime, activate, and sustain T cell responses across cancer types and are required for durable immunity against tumors[9,10]. Moreover, they hold a dominant role in extinguishing ongoing immune responses, making them among the most powerful regulators of antitumoral T cell responses.

[1]Department of Pathology, University of Michigan, Ann Arbor, MI, USA. [2]Michigan Center for Translational Pathology, University of Michigan, Ann Arbor, MI, USA. [3]Rogel Cancer Center, University of Michigan, Ann Arbor, MI, USA. [4]Department of Surgery, University of Michigan, Ann Arbor, MI, USA. [5]Center of Excellence for Cancer Immunology and Immunotherapy, University of Michigan, Ann Arbor, MI, USA. [6]Department of Computational Medicine & Bioinformatics, University of Michigan, Ann Arbor, MI, USA. [7]Howard Hughes Medical Institute, University of Michigan, Ann Arbor, MI, USA. [8]Department of Radiation Oncology, University of Michigan, Ann Arbor, MI, USA. [9]Department of Radiation Oncology Veterans Affairs Ann Arbor Healthcare System, Ann Arbor, MI, USA. [10]Department of Urology, University of Michigan, Ann Arbor, MI, USA. [11]Present address: Department of Pediatrics, University of California, San Francisco, CA, USA. [12]Present address: Feinberg School of Medicine, Northwestern University, Chicago, IL, USA. [13]Present address: Department of Pediatrics, Children's Hospital of Philadelphia, Philadelphia, PA, USA. ✉e-mail: wzou@umich.edu; arul@umich.edu

Though there are many elements in the tumor microenvironment that favor a pro or anti-cancer response, we now understand that the success of many approved cancer therapies ultimately depends on the sustained activation of TAA-specific CD8[+] T cells through DCs[9,11,12]. For example, the "DC-T cell axis" is required for response to immune checkpoint blockade (ICB) therapy. Loss of either cell type reduces ICB efficacy in preclinical models[13]. DC or T cell signatures are associated with ICB treatment response in cancer patients[3,14–16]. Studies have shown that certain systemic chemotherapy agents may enhance DC antigen presentation by enhancing tumor cell antigenicity[17,18]. However, studies reporting a direct effect on DCs are rare[19,20], with a few agents promoting a more "DC-like" phenotype in suppressive myeloid-derived cells[21]. Studies have focused on chemotherapy-induced activation of DCs as a mechanism to potentiate immunotherapy strategies, such as DC vaccines or ICB[13,21–24].

The contribution of DCs to targeted therapy efficacy, such as protein kinase inhibitors, is largely unknown. Unlike ICB or chemotherapy, these drugs target DNA damage responses or oncogenic signaling pathways preferentially utilized by cancer cells[25,26]. Targeting BRAF, EGFR, PDGRA, KIT, PIK3CA, ALK, MET, ROS1, ERBB, CDK12 and others has improved cancer care, however durable responses are uncommon[27,28]. Historically, the ability of the FDA-approved kinase inhibitors to interact with the immune system was ignored. More recently, preclinical studies have included characterization of the immune system, focusing on associations between treatment response and T cell profiles[29–34]. Efficacy of a receptor kinase inhibitor was shown to be dependent on T cells which served as a rationale for combination therapy with ICB[35,36]. However, these studies have not explored the contribution of DCs to those outcomes. As the number of clinical trials combining immunotherapy and protein kinase inhibitors continue to increase, it is imperative to understand whether they rely on DCs and modulate the DC-T cell axis.

In this study, we identified a DC-associated kinase, PIKfyve. Our data show that both genetic loss of *Pikfyve* in CD11c[+] cells and pharmacologic PIKfyve inhibition enhance DC function and activate the NF-κB pathway. Furthermore, loss of *Pikfyve* in CD11c[+] cells alone enhanced anti-tumor effect and ICB response in vivo. Finally, we show that treatment with a PIKfyve inhibitor potentiates vaccine-mediated tumor growth reduction in vivo. Together, these results suggest a role for targeting PIKfyve to alter DC phenotypes and DC-dependent therapies in the tumor microenvironment.

## Results

### DC PIKFYVE expression is associated with ICB efficacy
Protein kinases are important targets for cancer therapy. To explore the relevance of their gene targets in ICB-induced tumor immunity, we first identified the 25 most common and unique gene targets of Phase I/Phase II/FDA-approved kinase inhibitors offered in a commercial screening library[37–39] *(ALK, AURKA, BTK, CSF1R, EGFR, FGFR1, FLT1, FLT3, IGF1, IGFR1, IKBKB, JAK1, JAK2, KDR, KIT, MET, MTOR, NTRK1, PDGFRA, PIKFYVE, PTK2, RAF1, RET, SRC, and SYK).* We then explored the potential involvement of these 25 genes in treatment response in a clinical cohort of patients who received ICB and clinical bulk RNA-sequencing (RNA-seq) of their tumors at the University of Michigan as a part of their cancer treatment (Table 1)[40].

Seventeen percent of our cohort exhibited RECIST-defined[41] complete response (CR) to ICB (Fig. 1a). As expected, the CR group had the best overall survival in our cohort (Fig. 1b). Of the kinase inhibitor gene targets evaluated, high pre-treatment *PIKFYVE, KDR, NTRK1, IKBKB, FLT1, or FGFR1* expression was each associated with lower odds of achieving CR when controlling for cancer type, biopsy site, age at the time of treatment initiation, and ICB agent (Fig. 1c). Amongst these targets, we found that only high *PIKFYVE* expression was independently associated with worse progression-free survival (PFS) on treatment (Fig. 1d). Furthermore, a high pre-treatment CD8[+] T cell

**Table 1 | Demographics of a clinical cohort of patients treated with ICB**

| Percent (n) | | Hazard ratio [95% CI] | P value |
|---|---|---|---|
| *Age (time of sequencing)* | | | |
| Below median | 50.0% (n = 46) | 1 (reference level) | – |
| Above median | 50.0% (n = 46) | 0.55 [0.29, 1.03] | 0.06 |
| *Sex* | | | |
| Female | 44.5% (n = 41) | 1 (reference level) | – |
| Male | 55.5% (n = 51) | 0.92 [0.46, 1.86] | 0.82 |
| *Immune checkpoint blockade agent* | | | |
| Atezolizumab | 9.8% (n = 9) | 1 (reference level) | – |
| Nivolumab | 27.2% (n = 25) | 0.43 [0.13, 1.40] | 0.16 |
| Pembrolizumab | 59.7% (n = 55) | 0.58 [0.23, 1.44] | 0.24 |
| Combination | 3.3% (n = 3) | 0.43 [0.065, 2.92] | 0.39 |
| *Cancer type* | | | |
| Melanoma | 9.8% (n = 9) | 1 (reference level) | – |
| Bladder | 16.3% (n = 15) | 1.24 [0.39, 3.86] | 0.70 |
| Breast | 14.1% (n = 13) | 0.73 [0.22, 2.34] | 0.60 |
| Gastrointestinal | 3.3% (n = 3) | 1.20 [0.20, 7.02] | 0.83 |
| Head and Neck | 20.7% (n = 19) | 0.76 [0.28, 2.05] | 0.60 |
| Kidney | 6.5% (n = 6) | 0.27 [0.053, 1.38] | 0.11 |
| Lung | 5.4% (n = 5) | 0.10 [0.011, 0.92] | 0.042 |
| Lymphoma | 3.3% (n = 3) | 2.41 [0.53, 10.99] | 0.25 |
| Prostate | 12.0% (n = 11) | 1.16 [0.39, 3.49] | 0.77 |
| Sarcoma | 4.3% (n = 4) | 0.57 [0.13, 2.40] | 0.44 |
| Other | 4.3% (n = 4) | 1.97 [0.53, 7.30] | 0.30 |

Pre-treatment tumor bulk RNA-seq libraries from 92 patients who received ICB treatment and clinical sequencing at the University of Michigan were included in the analysis. Hazard ratios for overall survival are included with 95% confidence intervals. Two-sided p values were calculated from a multivariate cox proportional hazards model.

activation score[42,43] was associated with better overall survival, whereas a high PIKfyve score had the opposite effect (Fig. 1e). In a published prospective study[44], we found that pre-treatment PIKfyve scores were not different between patients with melanoma who were ICB-treatment-naïve ("NAÏVE") or had previously progressed on treatment ("PROG") (Supp. Fig. 1a). Interestingly, a high PIKfyve score was associated with worse overall survival in the PROG cohort when controlling for mutational subtype, stage of disease, and neoantigen load (Supp. Fig. 1b, c). Therefore, *PIKFYVE*, encoding a cytosolic lipid kinase[45–47], may be a therapeutically targetable gene with relevance in patients receiving ICB.

We then postulated that the correlation between *PIKFYVE* and ICB-associated outcomes in cancer patients may depend on cell type. We utilized published single cell RNA-seq (scRNA-seq) datasets to assess this hypothesis. We found that *PIKFYVE* was expressed in immune cells across multiple cancer types[48] (Supp. Fig. 1d–h). Importantly, in a study of patients with melanoma treated with ICB[49], pre-treatment expression of *PIKFYVE* in conventional DCs (cDC) was lower in responders (R) when compared to non-responders (NR) (Fig. 1f, g). There were no differences in *PIKFYVE* expression in any other immune cell type. In addition, a patient with endometrial cancer with CR[50] had lower expression of *PIKFYVE* in their cDCs compared to nonresponders (Supp. Fig. 1i, j). Together, these data nominate DCs as an immune cell in which *PIKFYVE* may play a significant role in cancer immunity and ICB-associated outcomes.

### PIKfyve suppresses function in CD11c+ cells
Given these unexpected findings, we sought to understand why patients with lower DC *PIKFYVE* expression in tumors were inclined

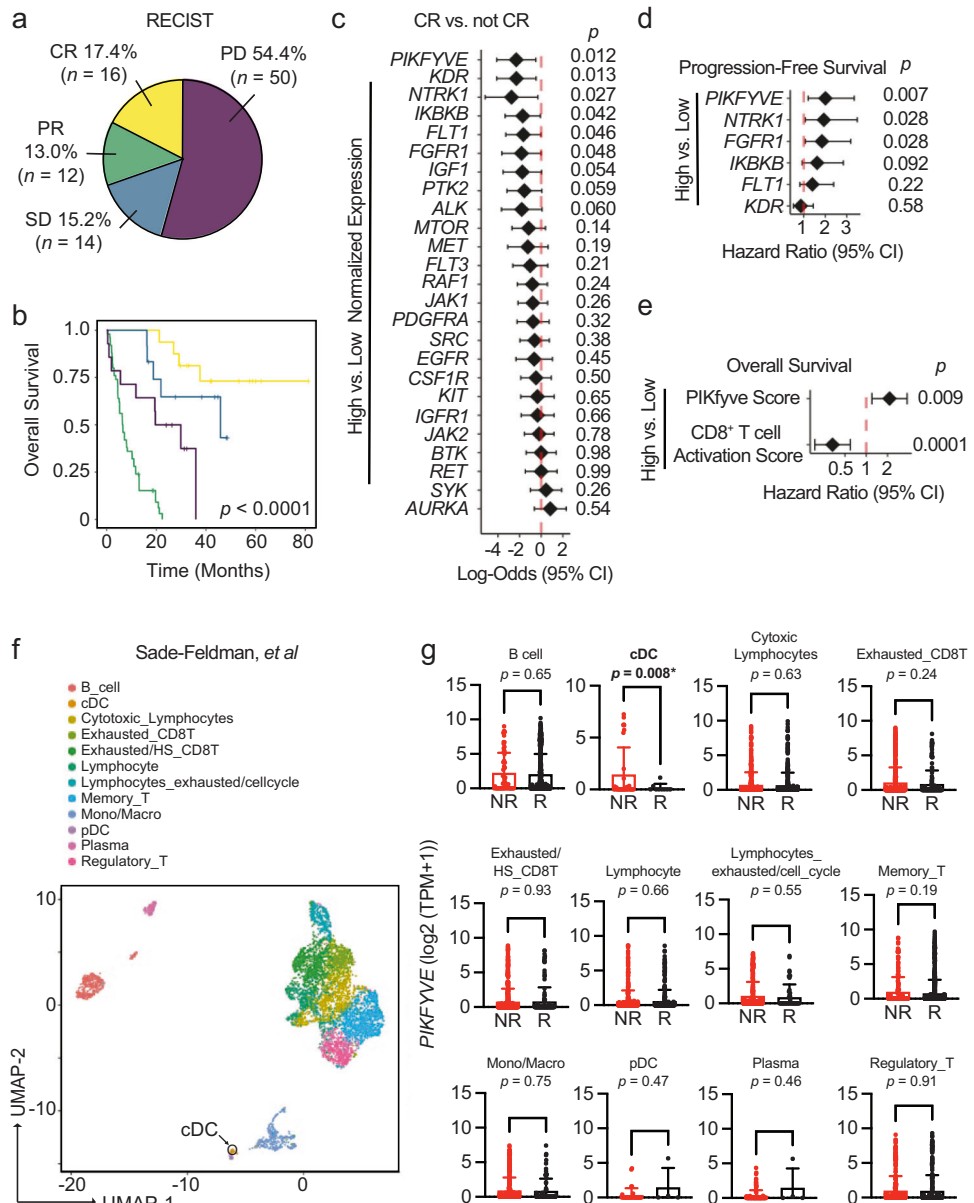

**Fig. 1 | DC *PIKFYVE* expression is associated with ICB efficacy. a** Pie chart of number and percentages of RECIST-defined response to treatment of patients treated with immune checkpoint blockade (ICB) at the University of Michigan, Ann Arbor (*n* = 92). CR complete response, PR partial response, SD stable disease, PD progressive disease. **b** Kaplan–Meier curves of overall survival of patients treated with ICB, by RECIST-defined treatment response. *P* value of 3.81 × 10⁻¹³ is deter-mined by log-rank test. **c** Forest plot of log-odds of having complete response (CR) or not CR for 25 common Phase I/Phase II/FDA-approved drug target genes. Data from bulk RNA-seq from 92 patients are plotted as log-odds with 95% confidence intervals. *P* values are determined by multivariate logistic regression controlling for cancer type, biopsy, ICB agent, and age at initiation of ICB treatment. **d** Forest plot of hazard ratios for progression-free survival of patients treated with ICB, by high or low gene expression for candidate drug target genes. Data from bulk RNA-seq hazard ratios with 95% confidence intervals. *P* values are determined by

multivariate cox proportional hazards model controlling for cancer type, ICB agent, and age at initiation of ICB treatment. **e** Forest plot of hazard ratios for overall survival of patients treated with ICB, by high or low gene expression for PIKfyve score and CD8⁺ T cell activation score. Data from bulk RNA-seq hazard ratios with 95% confidence intervals. *P* values are determined by multivariate cox proportional hazards model controlling for cancer type, ICB agent, and age at initiation of ICB treatment. **f** t-SNE plot of 5928 immune cells in pre-treatment tumors from patients with melanoma from scRNA-seq data[49]. The black arrow indicates the cDC popu-lation. **g** *PIKFYVE* expression (log2 (TPM + 1)) across immune cell types in non-responders ("NR" including SD and PD response) or responders ("R" including PR and CR response) to ICB treatment. Data plotted are mean ± s.d. from scRNA-seq data of 5928 immune cells[49]. *P* value determined by student *t*-test with Welch's correction. All *P* values are two-sided without correction for multiple comparisons. Source data are provided as a Source data file.

towards better clinical outcomes following ICB therapy. We hypothe-sized that PIKfyve may have a direct and functional role in modulating DCs and DC-mediated immune response against tumors. To this end, we performed genetic and pharmacologic manipulation of PIKfyve in DCs to gain a comprehensive and accurate portrayal of its functional role in this context.

First, we treated non-tumor-bearing, wild-type mice with apili-mod, a potent and specific PIKfyve inhibitor that has been studied in many cell types and evaluated in Phase II clinical trials[35,51–60]. Next, we performed a global assessment of immune cells enriched from spleens. We discovered increased proportions of CD69⁺CD8⁺ T cells in apilimod-treated mice compared to vehicle (Supp. Fig. 2a, b). There

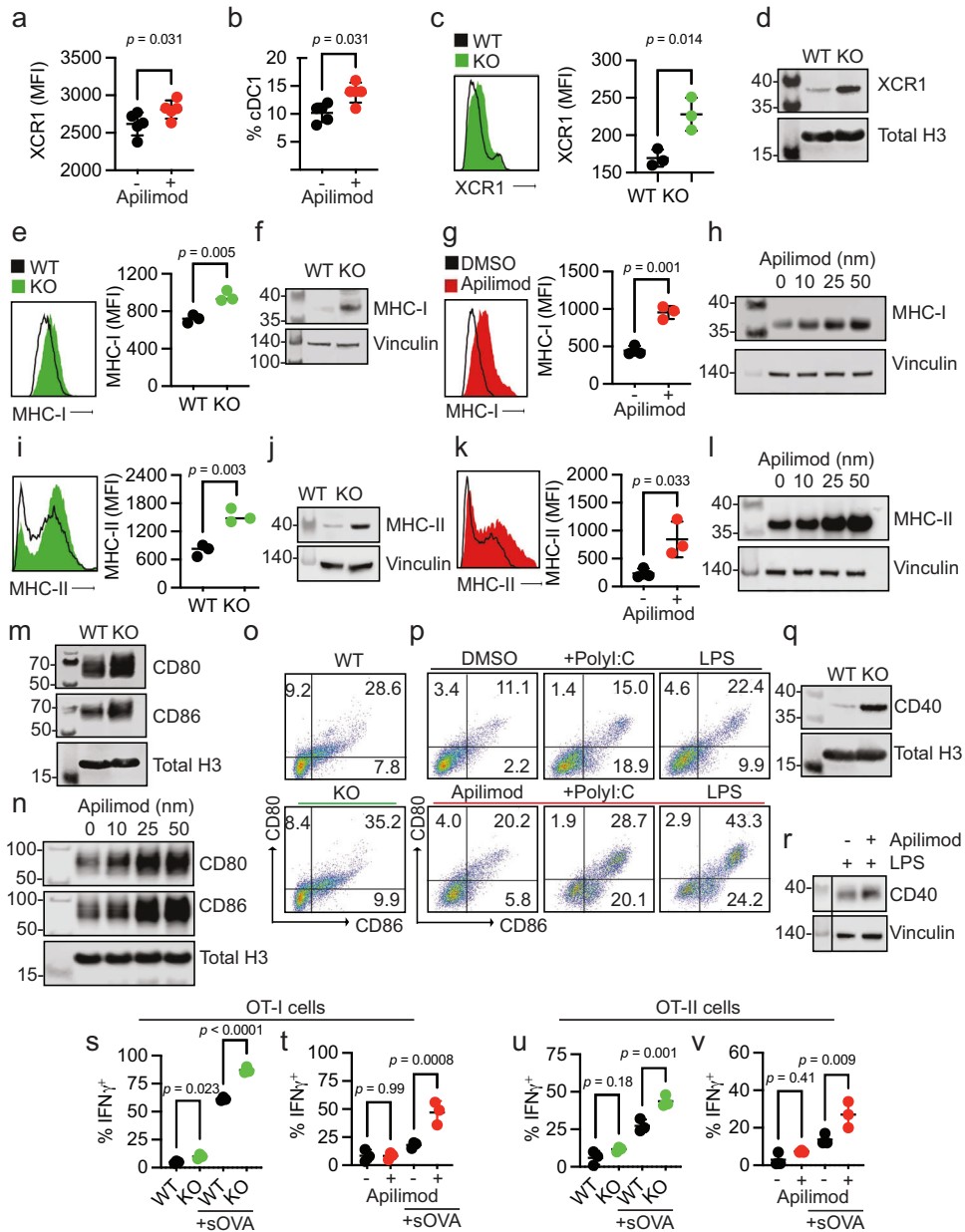

**Fig. 2 | PIKfyve suppresses function in CD11c[+] cells. a** Median fluorescent intensity (MFI) of XCR1 in splenic CD45[+] cells and **b** percentage of splenic cDC1s treated with vehicle or apilimod ($n = 5$ per group). **c** MFI of XCR1 on WT or KO cDCs. **d** Immunoblot of XCR1 in *Pikfyve* WT versus KO cDCs. Total histone H3 serves as loading control. **e** MFI and **f** immunoblot of MHC-I in *Pikfyve* WT or KO cDCs. **g** MFI and **h** immunoblot of total MHC-I in DMSO or apilimod-treated cDCs after 20 h. Vinculin serves as loading control. **i** MFI and **j** immunoblot of MHC-II in *Pikfyve* WT or KO cDCs. **k** MFI and **l** immunoblot of MHC-II in DMSO or apilimod-treated cDCs after 20 h. Vinculin serves as loading control. Immunoblots of CD80 and CD86 **m** in *Pikfyve* WT or KO and **n** in DMSO or apilimod-treated cDCs. Total histone H3 shown as a representative loading control. Dot plots of CD80 and CD86 on **o** *Pikfyve* WT or KO cDCs (representative of three experiments) and **p** in DMSO or apilimod-treated cDCs ± PolyI:C (50 μg/ml) or lipopolysaccharide (LPS, 50 ng/ml) treated for 20 h

(representative of two experiments). **q** Immunoblot of CD40 in *Pikfyve* WT or KO cDCs. Total H3 serves as the loading control. **r** Immunoblot of CD40 in DMSO or apilimod-treated cDCs treated with LPS (50 ng/ml) after 20 h. Vinculin serves as the loading control. Vertical line denotes separation between molecular weights and experimental lanes. Percentage of IFNγ[+] OT-I cells co-cultured with **s** *Pikfyve* WT or KO or **t** DMSO or apilimod pre-treated cDCs ± sOVA and OT-II cells co-cultured with **u** WT or KO or **v** DMSO or apilimod pre-treated cDCs ± sOVA ($n = 3$ biological replicates). Data are mean ± s.d. *P* value from ANOVA (Sidak adjustment for multiple comparisons). All *P* values are two-sided. All MFI data are mean ± s.d with *P* value determined by student *t*-test ($n = 3$ biological replicates). All immunoblots are representative of two experiments. Source data are provided as a Source data file which includes additional loading controls for (**m**, **n**) and image for (**r**).

---

were no changes in the percentage of terminally differentiated, memory or exhausted CD8[+] T cells between groups (Supp. Fig. 2c–g). We also assessed myeloid lineages (Supp. Fig. 2h). We found no change in total surface expression of CD11c, F4/80, or CD11b (Supp. Fig. 2i–k). Interestingly, there was an increase in the total surface expression of XCR1 with apilimod treatment (Fig. 2a). Within the cDC subset, there was an increase in the relative percentage of cDC1s (Fig. 2b) versus

cDC2 cells (Supp. Fig. 2l) with apilimod treatment. This warranted further study as cDC1s are a cDC subtype which expresses XCR1 and is involved in cross-presentation and CD8[+] T cell-mediated anti-tumor responses[61–63]. An increase in total surface expression of XCR1 was also observed in cultured cDCs treated with apilimod compared to dimethyl sulfoxide (DMSO) (Supp. Fig. 3a, b). To confirm that these changes were due to loss of functional PIKfyve, we bred *Pikfyve*[f/f] ("WT") mice

with *Itgax*^[Tg/0] ("CD11c-Cre") mice to specifically and conditionally delete *Pikfyve* in CD11c⁺ cells, which are predominantly DCs ("KO"). We cultured cDCs isolated from WT and KO mice in vitro and validated loss of PIKfyve expression (Supp. Fig. 3c). Intriguingly, we found that surface (Fig. 2c) and total protein (Fig. 2d) expression of XCR1 was also increased in cDCs from *Pikfyve* KO versus WT mice.

Given this evidence that *Pikfyve* loss could alter cDC state, we broadly assessed phenotypic changes in cDCs with PIKfyve ablation. We found that surface (Fig. 2e) and total protein (Fig. 2f) expression of MHC-I (H2-kb, H2-kd) were significantly increased in KO cDCs when compared to those from WT mice. To determine if these findings were phenocopied by pharmacologic PIKfyve inhibition, we treated cDCs from WT mice with apilimod. Apilimod-treated cDCs showed increased surface and total protein expression of MHC-I when compared to control (Fig. 2g-h). Similarly, surface (Fig. 2i) and total protein expression (Fig. 2j) of MHC-II were increased in the KO versus WT cDCs. Furthermore, apilimod-treated cDCs had increased surface and total protein expression of MHC-II (Fig. 2k-l).

To understand whether this increase in MHCI/II expression occurred in isolation or represented a change in overall DC maturation, we examined the expression of co-stimulatory molecules. Total protein levels of CD80 and CD86 expression were increased in *Pikfyve* KO versus WT cDCs (Fig. 2m). Apilimod treatment induced CD80 and CD86 expression in WT DCs in a dose-dependent manner (Fig. 2n). The percentage of CD80⁺CD86⁺ cDCs was also increased with *Pikfyve* KO (Fig. 2o, Supp. Fig. 3d) compared to WT. This increase in CD80⁺CD86⁺ cDCs was mirrored in apilimod-treated cells alone and in combination with PolyI:C and LPS (Fig. 2p). Furthermore, the total expression of CD40 was increased in *Pikfyve* KO versus WT cDCs (Fig. 2q). Addition of apilimod to LPS treatment also increased total expression of CD40 when compared to DMSO-treated cDCs (Fig. 2r).

As genetic and pharmacologic ablation of PIKfyve affected MHC and costimulatory molecules, we hypothesized that PIKfyve could, consequently, alter the ability of cDCs to present antigens and subsequently activate antigen-specific T cells. To test this possibility, we utilized the ovalbumin (OVA) and OT-I/OT-II cell systems[43,64,65]. When cultured with SIINFEKL peptide (pOVA), *Pikfyve* KO cDCs had greater total intensity and percentage of cells with H2-kb-SIINFEKL surface expression compared to WT cDCs (Supp. Fig. 3e, f). Similarly, apilimod-treated cDCs had greater total intensity and percentage of cells with H2-kb-SIINFEKL surface expression on cDCs compared to DMSO-treated cDCs (Supp. Fig. 3g, h). In addition, we observed that peptide antigen-loaded *Pikfyve* KO cDCs induced a greater percentage of IFNγ⁺ and granzyme B⁺ OT-I cells compared to WT (Supp. Fig. 4a–c). This increase in T cell activation was also seen with apilimod treatment compared to DMSO (Supp. Fig. 4d, e). Finally, we loaded cDCs with soluble OVA protein (sOVA) which were then co-cultured with OT-I and OT-II cells. There were higher percentages of Ki67⁺ (Supp. Fig. 4f) and IFNγ⁺ (Fig. 2s) OT-I cells following co-culture with sOVA-loaded *Pikfyve* KO cDCs compared to WT cDCs. These increases were also demonstrated in OT-I cells co-cultured with apilimod-treated cDCs compared to DMSO (Fig. 2t, Supp. Fig. 4g). In addition, there were higher percentages of Ki67⁺ and IFNγ⁺ OT-II cells following co-culture with Pikfyve KO and apilimod-treated cDCs when compared to their respective controls (Supp. Fig. 4h–i, Fig. 2u, v). Together, these data indicate that PIKfyve alters DC state and negatively controls DC maturation and function.

## PIKfyve suppresses NF-κB activation in DCs

To explore the mechanism through which PIKfyve suppresses DCs, we conducted RNA-seq studies. First, we performed gene set enrichment analysis on genes differentially expressed between *Pikfyve* KO versus WT cDCs (Supp. Data 1). Review of MSigDB curated gene sets ("C2") revealed that a gene signature of enhanced DC maturation was positively enriched in *Pikfyve* KO cDCs (Supp. Fig. 5a, Supp. Data 2)

consistent with our phenotypic and functional characterization of these cells.

Furthermore, examination of MSigDB hallmark gene sets ("H") revealed positive enrichment of the "TNF_SIGNALING_VIA_NFκB" gene set in *Pikfyve* KO versus WT cDCs (Supp. Fig. 5b, Supp. Data 3). This was intriguing as NF-κB is known to be an essential transcription factor for driving the overall maturation and acute activation of DCs[66–69]. Given our findings, we hypothesized that a more general signature of NF-κB downstream gene targets would be affected in *Pikfyve* KO cDCs. Thus, we curated a list of validated NF-κB gene targets from commercial assays (Supp. Data 4). In a posteriori analysis, we observed that this signature representing direct activation of downstream NF-κB genes was positively enriched in *Pikfyve* KO versus WT cDCs (Fig. 3a).

To corroborate these data suggesting NF-κB regulation by PIKfyve, we treated WT cDCs with apilimod to identify differentially expressed genes at timepoints coinciding with early DC activation (Supp. Data 5 and 6). Examination of MSigDB hallmark gene sets revealed positive enrichment of "TNF_SIGNALING_VIA_NFκB" in apilimod-treated cDCs at 3 and 8 h when compared to DMSO (Supp. Fig. 5c, d, Supp. Data 7 and 8). Importantly, direct NF-κB gene targets were positively enriched in apilimod-treated cDCs at these early timepoints, thus substantiating our findings from the genetic model (Fig. 3b, c). Interestingly, apilimod treatment increased *Il12b* transcripts (Supp. Fig. 5e, Supp. Data 5 and 6) in addition to the secretion of IL-12p40 (Supp. Fig. 5f) and IL-12p70 (Supp. Fig. 5g). Finally, we confirmed that both genetic loss and pharmacological inhibition of PIKfyve increased relative protein levels of p-NF-κB to NF-κB (Fig. 3d).

NF-κB is regulated by dynamic, cascading changes in a well-characterized system of upstream cytosolic regulatory proteins that culminate in an altered transcriptional landscape[70,71]. Interestingly, neither genetic nor pharmacological PIKfyve ablation changed levels of IκB-β, but PIKfyve loss or inhibition did increase levels of p-IκB-α relative to IκB-α, which would allow for increased NF-κB phosphorylation and activation (Fig. 3e). To understand if this regulation extended further upstream, we also investigated the canonical and alternate/non-canonical regulators of the IκB kinase complex. Abundance of IKK-α and IKK-β were unchanged with PIKfyve ablation (Fig. 3f). Interestingly, the abundance of IKK-ε, p-TBK-1, and TBK-1 were increased with genetic and pharmacological PIKfyve ablation (Fig. 3g).

These findings motivated further exploration of how PIKfyve may alter NF-κB activation. Intriguingly, *Sqstm1* emerged as the most significantly upregulated gene in apilimod-treated cDCs when compared to DMSO (Fig. 3h, Supp. Data 6). It was also differentially expressed in KO versus WT cDCs (Supp. Data 1). *Sqstm1* is a promising mechanistic candidate in PIKfyve regulation as it is involved in vesicular trafficking, autophagy and lysosome and proteasome degradation[72,73]. Importantly, *Sqstm1* can regulate NF-κB activation under various conditions[72,74–76] and is known to interact with TBK-1[72,73,77–80].

Utilizing the TCGA Pan-Cancer data, we found that high *SQSTM1* was associated with better overall survival in patients (Supp. Fig. 6a). We also found that high *SQSTM1* negatively correlated with PIKfyve scores (Fig. 3i). Furthermore, patients with high NF-κB gene target scores were more likely to have high *SQSTM1* expression (Fig. 3j). Recently, Ghislat et al.[67] demonstrated that NF-κB pathway activation may direct maturation states within the mouse cDC1 subset. We discovered *Sqstm1* expression was higher in mouse cDC1[81] when compared to most other myeloid lineages, including cDC2 (Supp. Fig. 6b). We investigated this further in the patient data. As expected, a published[67] "cDC1 maturation score" positively correlated with NF-κB gene targets in TCGA Pan-Cancer patient samples (Supp. Fig. 6c). Moreover, patients who had higher cDC1 maturation scores were more likely to have high *SQSTM1* expression (Fig. 3k). Collectively, these data suggest that PIKfyve mediates suppression of DC transcriptional maturation programs through the alternate/non-canonical NF-κB regulatory pathway.

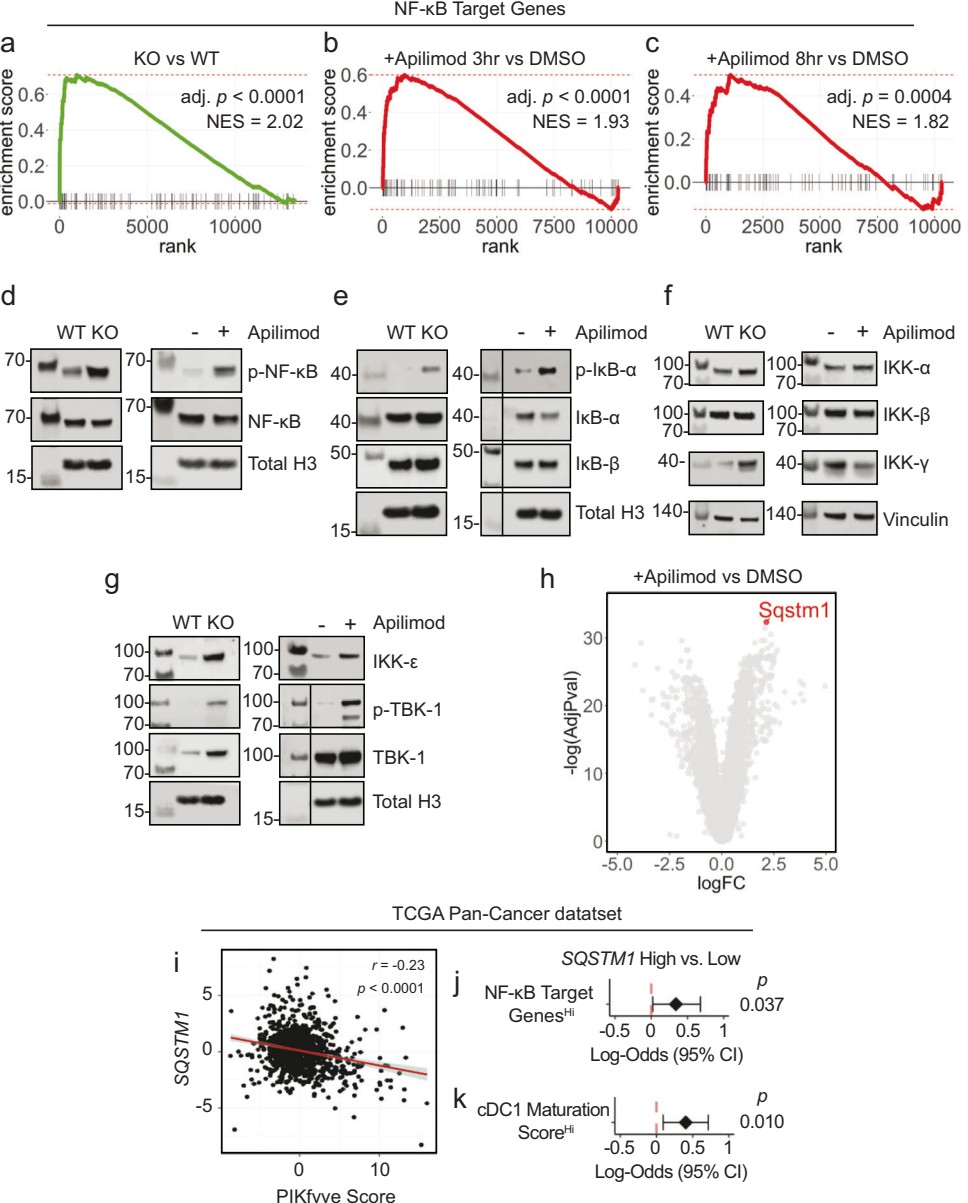

**Fig. 3 | PIKfyve suppresses NF-κB activation in DCs. a** Enrichment plots of NF-κB target genes in *Pikfyve* KO or WT cells on culture day 9 ($p = 2.37 \times 10^{-7}$). Enrichment plots of NF-κB target genes in DMSO or apilimod-treated cDCs treated for **b** 3 h ($p = 1.73 \times 10^{-5}$) or **c** 8 h on culture day 6. Adjusted *p* values accounting for multiple comparisons were calculated using fast pre-ranked gene set enrichment analysis. **d** Immunoblots of p-NF-κB-p65 and total NF-κB-p65 (left) in *Pikfyve* WT or KO and (right) in DMSO or apilimod-treated cDC lysates. **e** Immunoblots of p-IκB-α, IκB-α, and IκB-β (left) in *Pikfyve* WT or KO and (right) in DMSO or apilimod-treated cDC lysates. **f** Immunoblots of IKK-α, IKK-β, and IKK-γ (left) in *Pikfyve* WT or KO and (right) in DMSO or apilimod-treated cDC lysates. **g** Immunoblots of IKK-ε, p-TBK-1, and TBK-1 (left) in *Pikfyve* WT vs. KO and (right) in DMSO or apilimod-treated cDC lysates. WT and KO lysates were collected on culture day 9. DMSO and apilimod lysates were collected after 20 h of treatment on culture day 6. Total histone H3 or vinculin serve as representative loading controls. Vertical line denotes separation

between molecular weights and experimental lanes. Additional loading controls for **d–g** and images for (**e, g**) are included in the Source Data. Images are representative of two experiments. **h** *Sqstm1* shown on volcano plot of differentially expressed genes in apilimod or DMSO-treated cDCs treated for 8 h on culture day 6. Adjusted *p* values accounting for multiple comparisons were calculated from linear models. **i** *SQSTM1* expression and PIKfyve score in TCGA Pan-Cancer RNA-seq dataset. Correlation coefficient and *p* value calculated using Pearson's product-moment correlation. Fitted linear line with shaded 95% confidence interval is plotted. **j** Forest plot of log-odds of high or low *SQSTM1* for patients with high or low NF-κB gene targets expression or **k** high vs. low cDC1 maturation score[67]. Data from bulk RNA-seq plotted are log-odds with 95% confidence intervals. *P* values are determined by multivariate logistic regression controlling for cancer type. All *P* values are two-sided. Source data are provided as a Source data file.

## Pikfyve loss in CD11c⁺ cells enhances anti-tumor immunity in vivo

Since the above results showed that PIKfyve could fundamentally transform cDCs, we examined whether these alterations had any effect on a disease setting. We hypothesized that the loss of *Pikfyve* in CD11c⁺ cells could attenuate tumor growth in syngeneic mouse models of cancer. Subcutaneous MC38 tumors injected into *Pikfyve* KO mice

manifested reduced tumor growth (Fig. 4a) and tumor weights (Supp. Fig. 7a) compared to WT mice. Furthermore, this inhibitory effect on tumor growth and terminal tumor weights was also seen in KO mice bearing MCA-205 (Fig. 4b, Supp. Fig. 7b) and B16F10 (Fig. 4c, Supp. Fig. 7c) tumors. We additionally explored whether loss of *Pikfyve* in CD11c⁺ cells could alter response to anti-PD-1 therapy in the ICB-resistant B16F10 model. Tumor growth inhibition with anti-PD-1

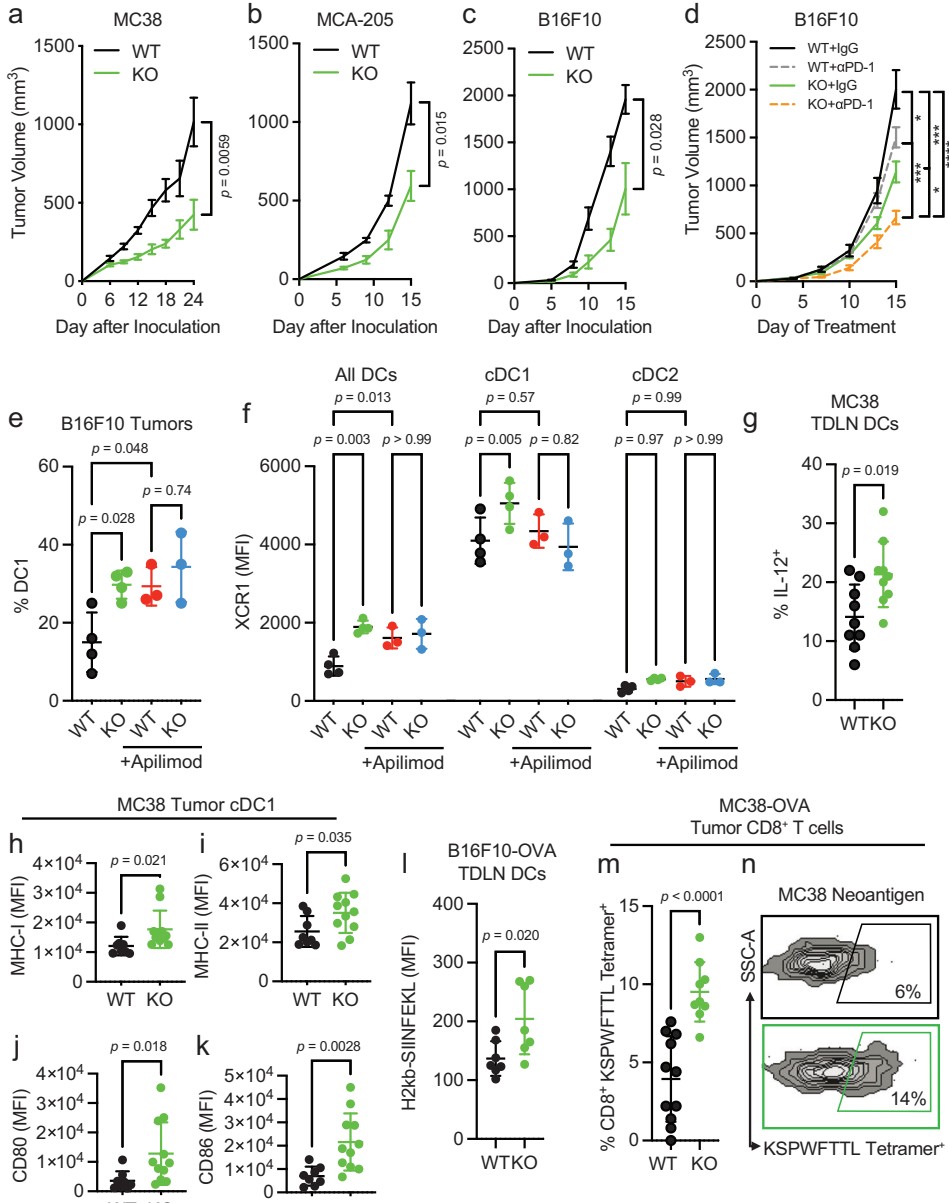

**Fig. 4 | *Pikfyve* loss in CD11c⁺ cells enhances anti-tumor immunity in vivo.** Subcutaneous tumor volumes measured in mm³. **a** Volume of MC38 tumors in *Pikfyve* KO mice (*n* = 8) or WT mice (*n* = 7) on Day 24. **b** Volume of MCA-205 tumors in *Pikfyve* KO mice (*n* = 6) or WT mice (*n* = 6) on Day 15. **c** Volume of B16F10 tumors in *Pikfyve* KO mice (*n* = 8) or WT mice (*n* = 8) on Day 15. Data plotted are mean ± s.e.m. *P* values are determined by Mann−Whitney *U* test. **d** Volume of B16F10 tumors in *Pikfyve* KO mice or WT mice treated with anti-PD1 antibody or isotype control (*n* = 8 per group) on Day 15. Data plotted are mean ± s.e.m. *P* value determined by ANOVA after post-hoc Tukey adjustment for multiple comparisons. **e** % cDC1 of all DCs and **f** XCR1 MFI by all DC, cDC1 or cDC2 subset in *Pikfyve* KO mice or WT mice bearing B16F10 tumors treated with vehicle or apilimod. Bilateral subcutaneous tumors from each mouse (*n* = 4 per group) were combined for analysis. Data plotted are mean ± s.d. *P* value is determined by ANOVA after post-hoc Tukey adjustment for multiple comparisons on day 6 of treatment. **g** Percentage of intracellular IL-12⁺ expression in WT (*n* = 9) or KO (*n* = 9) cDCs from tumor-draining lymph nodes of MC38 tumors after 20 h of LPS (20 ng/ml). **h** MHC-I, **i** MHC-II, **j** CD80 and (**k**) CD86 MFI of cDC1 subset in tumors of *Pikfyve* KO (*n* = 11) or WT mice (*n* = 8) on day 10. Data plotted are mean ± s.d. *P* value determined by student *t*-test with Welch's correction. **l** MFI of surface H2-kb-SIINFEKL in WT (*n* = 6) vs. KO (*n* = 6) cDCs from B16F10-OVA tumor-draining lymph nodes. Data plotted are mean ± s.d. *P* value determined by Mann−Whitney *U* test. **m** Percentage and **n** dot plots of KSPWFTTL Tetramer⁺ CD8⁺ T cells isolated from MC38-OVA WT (*n* = 11) or KO (*n* = 9) tumors. Data plotted are mean ± s.d. *P* value determined by student *t*-test with Welch's correction. All *P* values are two-sided. Source data are provided as a Source data file.

therapy was enhanced when comparing endpoint tumor volume in *Pikfyve* KO mice to wild type mice (Fig. 4d).

We then assessed the effects of PIKfyve inhibitor treatment on tumor infiltrating DCs and T cells isolated from DC-selective *Pikfyve* KO versus WT mice in multiple tumor models. There was an increase in the percentage of cDC1s in KO versus WT mice in B16F10 tumors (Fig. 4e, Supp. Fig. 7d). Importantly, treatment with apilimod increased the

percentage of cDC1s amongst the WT mice but resulted in no additional increase in the KO mice. Notably, these differences were the same when evaluating total surface XCR1 expression between groups (Fig. 4f). There were no changes in the percentage of cDC2s (Supp. Fig. 7e, f) or total surface SIRP1α expression (Supp. Fig. 7g) between any groups. Furthermore, we found an increased proportion of IL-12-expressing DCs in MC38 tumors from *Pikfve* KO versus WT mice

(Fig. 4g). When comparing *Pikfyve* KO versus WT tumors, MHC-I and cDC maturation markers (MHC-II, CD80, and CD86) were increased in the cDC1 subset (Fig. 4h–k) when compared to cDC2 subset (Supp. Fig. 7h–k) or all DCs (Supp. Fig. 7l–o). We also assessed tumor infiltrating CD8[+] and CD4[+] T cells from MC38 tumors. In the *Pikfyve* KO mice, there was a higher percentage of CXCR3[+] (Supp. Fig. 8a, b) and CD69[+] (Supp. Fig. 8c, d) effector T cells and a higher percentage of CD44[Hi] effector memory T cells (Supp. Fig. 8e, f) without changes in CD62L[+] naïve T cells (Supp. Fig. 8g, h). Interestingly, though there was a higher proportion of KLRG1[+] (Supp. Fig. 8i, j) CD8[+] and CD4[+] T cells in the KO group, PD-1 (Supp. Fig. 8k, l) and TIM-3 (Supp. Fig. 8m, n) were only increased in the CD4[+] T cells. Hence, these data showed that *Pikfyve* loss remodeled DC phenotype and T cell function in tumors in vivo.

We additionally investigated whether the immune response was altered with *Pikfyve* loss in CD11c[+] cells. As expected, the growth of B16F10-OVA tumors (Supp. Fig. 9a–c) was decreased when inoculated in KO versus WT mice. There was no difference in the percentage of CD11c[+]MHC-II[+] DCs isolated from tumor-draining lymph nodes (TDLNs) compared to all CD45[+] cells (Supp. Fig. 9d). Importantly, overall H2-kb-SIINFEKL expression on the surface of DCs from the KO mice was higher than those from the WT (Fig. 4l). Concordantly, the percentage of H2kb-SIINFEKL[+] DCs was increased in the KO mice (Supp. Fig. 9e). When comparing the littermate pairs of WT and KO mice, there was a marginal increase in the percentage of OVA antigen-specific (SIINFEKL) tetramer[+] CD8[+] T cells isolated from B16F10-OVA tumors from each pair (Supp. Fig. 9f–h). In the MC38-OVA model, there was a higher percentage of SIINFEKL tetramer[+] CD8[+] T cells isolated from tumors (Supp. Fig. 9i, j). MC38-OVA tumor had a higher proportion of MC38 tumor neoantigen-specific (KSPWFTTL) tetramer[+] CD8[+] T cells (Fig. 4m, n) demonstrating that *Pikfyve* loss enhanced DC-mediated antigen presentation and priming of antigen-specific CD8[+] T cells in an in vivo tumor model. Combined, these data show that PIKfyve in DCs can modulate tumor outcomes in vivo.

## Immune effects of PIKfyve are DC-dependent

Though PIKfyve inhibitors have shown anti-tumor effects in various preclinical cancer models[35,56,60,82–84], it is unclear if the activity of DCs directly contributes to its therapeutic effect. Our combined genetic and pharmacologic in vivo experiments showed no additional change in *Pikfyve* KO DCs with the addition of PIKfyve inhibitor treatment. Therefore, we further hypothesized that therapeutic PIKfyve inhibition required the presence of DCs to exert its full anti-tumor effect in vivo. To this end, we inoculated WT mice with subcutaneous MC38 tumors. In vivo treatment with apilimod reduced MC38 tumor growth compared to vehicle (Fig. 5a) and increased the percentage of intratumoral IFNγ[+]CD8[+] T cells (Fig. 5b, Supp. Fig. 10a). The efficacy of apilimod was lost in immune-deficient NSG mice (Supp. Fig. 10b). To test whether drug efficacy required DCs, we used *Batf3[−/−]* mice[61–63] which have loss of cDC1s. Importantly, loss of apilimod efficacy remained when comparing *Batf3[−/−]* mice to immune-competent, WT mice (Fig. 5c).

We also investigated the importance of functional DC and T cell signaling pathways in apilimod treatment efficacy. In MC38 tumor-bearing mice, we observed that the efficacy of apilimod was reduced when mice were treated with neutralizing monoclonal antibodies against IL-12[13] (Supp. Fig. 10c) or IFNγ[35,42] (Supp. Fig. 10d) compared to isotype controls. As IL-12 is often expressed by DCs and IFNγ by T cells, the data suggests that in vivo efficacy of PIKfyve inhibition requires the presence of functional DC signaling and intact DC and T cell signaling pathways. To further validate this possibility in an additional tumor model, we inoculated WT, non-transgenic mice with subcutaneous B16F10 tumors. Treatment with apilimod in vivo reduced tumor growth compared to vehicle in this model (Supp. Fig. 10e). Again, efficacy of apilimod was lost when B16F10 tumors were inoculated into NSG (Supp. Fig. 10f) or *Batf3[−/−]* mice (Fig. 5d).

As vaccines are a DC-dependent immunotherapy strategy, we explored whether apilimod, as a DC-modulating and DC-dependent agent could potentially modulate cancer vaccine strategies. PolyI:C and TLR agonists are commonly used as vaccine adjuvants in different tumor vaccine models[85]. We treated DCs with PolyI:C or lipopolysaccharide (LPS) in the presence of apilimod in vitro. We found that PolyI:C, LPS, or apilimod stimulated MHC-I and MHC-II expression, and apilimod further enhanced this effect (Supp. Fig. 10g, h). In a proof-of-concept experiment, we explored whether PIKfyve inhibition, as a DC-maturing and DC-dependent therapy, could potentiate the effects of PolyI:C against tumor growth in vivo. First, we treated mice with apilimod or vehicle in addition to subcutaneous PolyI:C or water (control) for 21 days prior to inoculation with B16F10-OVA tumors (Fig. 5e). The combination of pre-treatment with apilimod and PolyI:C decreased subsequent tumor growth compared to control or either agent alone (Fig. 5f, Supp. Fig. 10i).

We then inoculated mice with B16F10-OVA tumors first and followed by combination therapy with vehicle or apilimod in addition to PolyI:C or water (Fig. 5g). Apilimod and PolyI:C alone resulted in modest tumor growth inhibition which was further potentiated by combination with both agents (Fig. 5h, Supp. Fig. 10j). These data suggest that PIKfyve inhibitors are clinically viable DC-dependent drugs and are promising candidates for combination therapy strategies with human-relevant DC-stimulating agents and adjuvants in cancer and other diseases.

## Discussion

In this work, we explored whether therapeutically actionable molecular signaling pathways regulate DCs. We revealed a previously unknown mechanism for PIKfyve in controlling DC state, maturation, and function through NF-κB regulation and demonstrated its ability to potentiate immunotherapy strategies through DCs. Thus, we generated translational insight into the potential of PIKfyve inhibitors for enhancing DC-dependent therapies in cancer, such as ICB and vaccines.

Our data comprehensively characterize a clinically viable protein kinase inhibitor strategy to enhance DC function. Though the investment into these cancer-targeted drugs has transformed cancer therapy outcomes for many patients, it is clear that further insight is required to understand how to achieve universally durable and curative responses[25,26,28,86]. With the historically recent acceptance of the essential role of immunity in cancer, new studies have shown that these drugs may modulate CD8[+] T cell responses and the efficacy of immunotherapy[29,31–36,87–89]. Notably, their impact on DCs remains largely unexplored. DCs have long been a desired target for cancer therapy given their essential role in T cell immunity[9–11]. At present, it is challenging to manipulate DCs directly due to their small numbers, diversity of roles across subsets, and lack of specific targets. Our data identified PIKfyve as a clinically-druggable molecular target in DCs. This work highlights the potential of targeting PIKfyve in DCs to improve T cell responses and immunotherapy.

Our study revealed a previously unexplored mechanistic association between PIKfyve, a lipid kinase that synthesizes phosphatidylinositol and regulates autophagy[45,47,51,52,90], and NF-κB, a transcription factor that is functionally critical for DC activation and maturation[66–70]. We demonstrate that NF-κB signaling activation through PIKfyve ablation selectively occurred through the alternate or non-canonical regulators of the IκB inhibitor complex, such as TBK-1[71,91,92]. Furthermore, we identified *SQSTM1*, which has been shown to regulate NF-κB signaling under a wide range of conditions, as a possible mechanistic linchpin[72,73,77–80]. As *SQSTM1* is a critical component of autophagic regulation through TBK-1[72,74–76], vesicle trafficking[72,73], and transcriptional regulation[73], it is a potential nexus between PIKfyve and alternate or non-canonical NF-κB regulation. In addition, we discovered that PIKfyve may preferentially alter cDC1s in tumor models. These

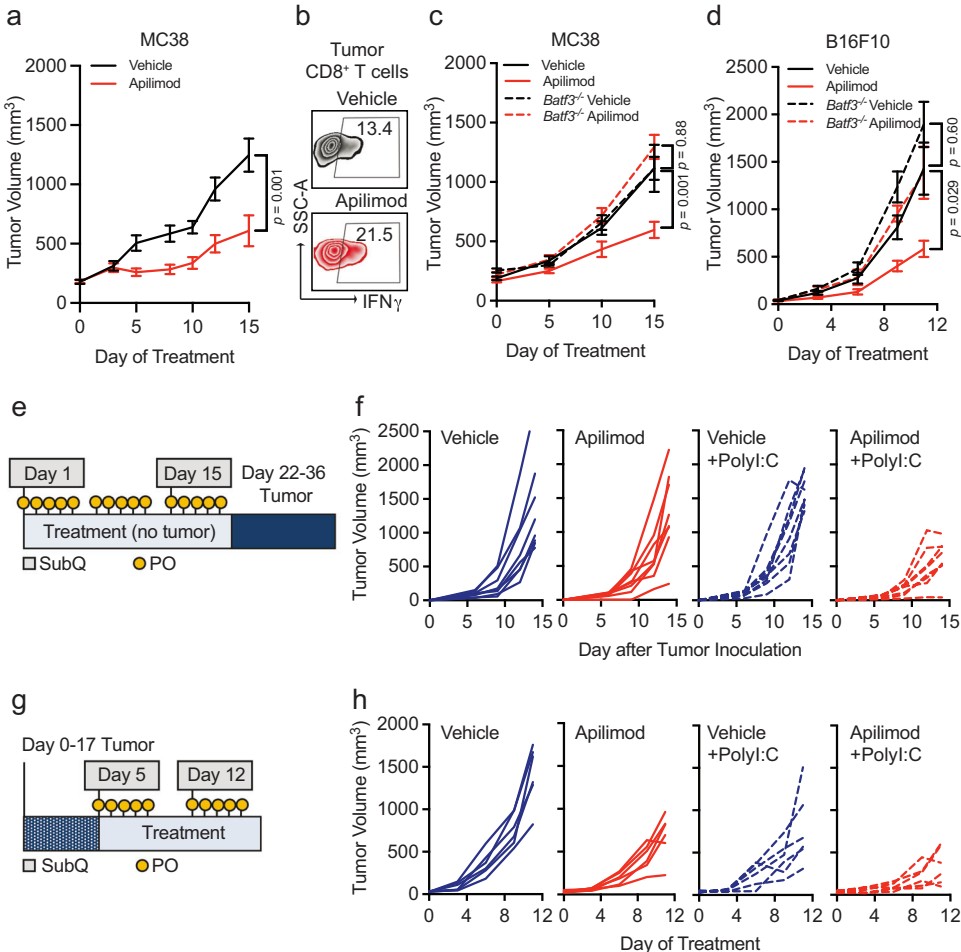

**Fig. 5 | Immune effects of PIKfyve are DC-dependent. a** Subcutaneous tumor volume of MC38 tumors (mm³) in mice treated with vehicle or apilimod (30 mg/kg × 5 days/week) (*n* = 10 tumors per group). Data plotted are mean ± s.e.m. *P* value determined by Mann–Whitney *U* test on day 15 of treatment. **b** Representative dot plots of percent IFNγ⁺ of CD8⁺ T cells in vehicle or apilimod-treated MC38 tumors. Bilateral subcutaneous tumors from each mouse were combined for analysis. **c** Subcutaneous tumor volume of MC38 tumors (mm³) in mice treated with vehicle versus apilimod (30 mg/kg × 5 days/week) in wild-type (*n* = 10 tumors per group) or *Batf3⁻/⁻* mice (*n* = 4 tumors per group). Data plotted are mean ± s.e.m. *P* value determined by ANOVA after post-hoc Tukey adjustment for multiple comparisons on day 15 of treatment. **d** Subcutaneous tumor volume of B16F10 tumors (mm³) in mice treated with vehicle or apilimod (30 mg/kg daily) in wild-type or *Batf3⁻/⁻* mice (*n* = 10 tumors per group). Data plotted are mean ± s.e.m. *P* value determined by ANOVA after post-hoc Tukey adjustment for multiple comparisons on day 9 of treatment. **e** Schematic of "vaccination" experiment demonstrating pre-treatment with oral (vehicle or apilimod) or subcutaneous (water or PolyI:C) reagents followed by B16F10-OVA tumor inoculation. **f** Individual growth curves of subcutaneous tumor volume of B16F10-OVA tumors (mm³) following 21 days of pre-treatment with vehicle versus apilimod (30 mg/kg daily) ± subcutaneous injection of water versus PolyI:C (100 µg on Day 1 and Day 14) (*n* = 8 tumors per group). **g** Schematic of "vaccine therapy" experiment demonstrating combination therapy treatment with oral (vehicle or apilimod) or subcutaneous (water or PolyI:C) reagents following B16F10-OVA tumor inoculation. **h** Individual growth curves of subcutaneous tumor volume of B16F10-OVA tumors (mm³) in mice treated with vehicle or apilimod (30 mg/kg daily) ± subcutaneous injection of water versus PolyI:C (100 µg once weekly) (*n* = 6 tumors per group). All *P* values are two-sided. Source data are provided as a Source data file.

results are in line with a recent study demonstrating that NF-κB signaling may be crucial for cDC1 maturation[67]. Our findings provide a rationale for further investigation of the potential for PIKfyve to regulate DC identity and lineage determination.

Ultimately, these data provide a strategy to transcriptionally and functionally overcome DC suppression in the tumor microenvironment through a lipid kinase. Given that PIKfyve suppresses DC function, we reason that PIKfyve inhibitors may be selected to treat cancers with high PIKfyve expression in tumors and DCs, thereby effectively targeting both tumors and the immune system. Conversely, caution may be required when considering using PIKfyve inhibitors in cancers driven by NF-κB signaling[93]. Thus, these fundamental insights into DC regulation may inform optimal, "precision medicine" treatment strategies for further evaluation of PIKfyve inhibitors in specific cancers.

To our knowledge, our results are the first to distinguish PIKfyve inhibition as a DC-dependent cancer therapy. Our preclinical studies demonstrate that PIKfyve inhibitors target DCs directly, require DCs and DC/T cell signaling to prevent tumor progression, and potentiate ICB therapy. This DC-dependent nature of PIKfyve inhibitor efficacy is rooted in its ability to mediate antigen presentation to T cells. Thus, our data provides a mechanistic rationale for PIKfyve inhibitor therapy in combination with ICB and invites further inquiry into ways to potentiate other DC-dependent strategies.

In addition to ICB, tumor vaccination is an approach for cancer prevention and therapy[94–99]. Many vaccination strategies involve adjuvants that directly activate DCs[85,98]. In proof-of-principle experiments, we found that PIKfyve inhibition could potentiate the antitumor effect of Poly I:C, a cancer vaccine adjuvant. As the results of cancer vaccine therapy have remained underwhelming[100], our data provides a rationale for further investigation of PIKfyve inhibitors to vaccinate against cancers and infectious diseases. As PIKfyve inhibition is now being clinically evaluated for efficacy against COVID-19[59,101], our

results substantiate the broad potential of PIKfyve inhibition as a DC-enhancing strategy across disease states.

## Methods

All research in this study was performed in compliance with all ethical regulations from the accredited Institutional Review Board at the University of Michigan, Ann Arbor and the Institutional Animal Care and Use Committee at the University of Michigan, Ann Arbor who approved the study protocols.

### Human studies

All clinical records in this study were obtained and utilized with the approval of Institutional Review Board for human subjects research and clinical trials (IRBMED) at the University of Michigan, Ann Arbor, and Michigan Medicine. All clinical records in this study were obtained with the approval of Institutional Review Boards, and the need for patient consent was waived following Institutional Review Board protocol review (HUM00146400, HUM00139259, HUM00163915, HUM00161860, HUM00046018). Patients who received ICB therapy were recruited through the University of Michigan Hospital, Ann Arbor, MI, USA. Those who were enrolled in the continuous comprehensive clinical sequencing program at the Michigan Center for Translational Pathology (MCTP MI-Oncoseq program)[40,42,43,102] and had bulk RNA-sequencing libraries of pre-treatment tumor were included in this analysis totalling 92 samples, including those from 41 females and 51 males. Sex was determined by self report and was not considered in the design of the analysis but was included as a covariate in the multivariate model. Gender was not reported. Overall survival times were determined from the initiation of therapy. Response to treatment was determined using RECIST1.1[41] criteria with patients meeting criteria for pseudoprogression (imRECIST criteria[103]) being excluded from analysis. Integrative clinical sequencing was performed using standard approved protocols in the MCTP Clinical Laboratory Improvement Amendments-compliant sequencing laboratory as previously described[40,102,104]. Total RNA purified using the AllPrep DNA/RNA/miRNA kit (Qiagen) was then sequenced using the exome-capture transcriptome platform on an Illumina HiSeq 2000 or HiSeq 2500 in paired-end mode. CRISP, the standard clinical RNA-Seq pipeline, was used to perform quality control, alignment, and expression quantification[105]. Tables of read counts were then transformed into fragments per kilobase of transcript per million mapped reads using the Bioconductor edgeR package in R[106].

### Gene signature score computation

We used normalized expression of genes to define CD8$^+$ T cell infiltration and activation (*CD8A, CXCL10, CXCL9, GZMA, GZMB, IFNG, PRF1, TBX21*), PIKfyve (*PIKFYVE, PIP4K2A*)[42,43], NF-κB gene targets and cDC1 maturation scores[67]. Scores were calculated by inverse-normal transformation of individual gene expression levels across the cohort followed by summation of inverse-normal values for each sample[42,43].

### Kinase inhibitor gene target selection

A catalog of kinase inhibitors of a commonly utilized commercial screening assay (Selleck Chem; Catalog L1800: https://www.selleckchem.com/screening/tyrosine-kinase-inhibitor-library.html)[37–39,107] was utilized to identify Phase I, Phase II, and FDA-approved drugs as previously described[37].

### Reagents

Apilimod was purchased from Selleck Chemicals and resuspended in dimethyl sulfoxide (DMSO) for in vitro studies and in 0.5% methylcellulose (vehicle) for in vivo experiments. Lipopolysaccharides (LPS) were purchased from Millipore Sigma (L2654) and resuspended in water. EndoFit Ovalbumin (vac-pova), H-2K$^b$-restricted ovalbumin MHC class I epitope (257–264) peptide (vac-sin), and polyI:C (HMW)

VacciGradeTM (vac-pic) were purchased from InvivoGen. EasySepTM Mouse CD8+ T cell isolation kit was purchased from Stemcell Technologies (19853).

### Bone marrow-derived conventional dendritic cells

Bone marrow was collected from the femurs and tibias of mice. Dendritic cells were generated with bone marrow cells cultured with Flt3L (200 ng/ml) in IMDM (Gibco: 12440-053) supplemented with 10% FBS. For genetic knock-out studies, cells were collected on days 8-10. For pharmacologic studies, cells were collected and treated on day 6. Primary cell cultures were tested to be mycoplasma-free in accordance with standard laboratory procedures at MCTP.

### Antigen presentation assays

For peptide-based antigen presentation assays, H-2Kb-restricted ovalbumin (OVA) peptide (100 ng/ml) was added to cDCs in vitro. CD8$^+$ T cells were enriched from the lymph node and spleens of OT-I mice. $1 \times 10^5$ OT-I T cells were then co-cultured with $1 \times 10^5$ cDCs previously cultured with H-2Kb-restricted OVA peptide for 12 h in 10% FBS medium with 55 μM β-ME and 5 ng/ml IL-2 in a 96-well plate. After 2 days, OT-I cells were collected and analyzed for cytokine expression.

For protein-based antigen presentation assays, CD3$^+$ T cells were enriched from the lymph nodes and spleen of OT-I or OT-II transgenic mice. $2 \times 10^5$ OT-I or OT-II cells were then co-cultured with $2 \times 10^5$ cDCs together with soluble ovalbumin (sOVA) (10 μg/ml) for 3 days in 10% FBS medium with 55 μM β-ME and 5 ng/ml IL-2 in a 96-well plate. After 3 days, OT-I or OT-II cells were collected and analyzed for cytokine expression. In experiments with drug treatment, cDCs were pretreated with DMSO or Apilimod (50 nM) for 4 h and washed prior to co-culture with T cells.

### Cell culture

MC38 cell line was acquired from Walter Storkus as previously described[65,108,109]. B16F10 and MCA-205 were purchased from ATCC. Ovalbumin-expressing B16F10 (B16F10-OVA) and MC38-OVA were established with pCI-neo-mOVA plasmid (Addgene plasmid #25099) and selected with 1 mg/ml of G418 for 2 weeks as previously described[42,43]. All cell lines were maintained at <70% in culture and tested for mycoplasma contamination every 2 weeks according to MCTP standard protocols.

### Flow cytometry analysis (FACS)

Cultured dendritic cells were collected and prepared as single cell suspensions. Single cell suspensions of mononuclear cells were isolated from bilateral tumors combined for each mouse with mechanical disassociation followed by Ficoll separation. Single cell suspensions of mononuclear cells were mechanically disassociated from bilateral tumor-draining lymph nodes.

Surface staining was performed by adding antibodies to single cell suspensions in MACS buffer (PBS, 2% FBS, 1 mM EDTA) for 30 min. For cytokine staining, cells were stimulated with phorbol myristate acetate (5 ng/ml), ionomycin (500 ng/ml), brefeldin A, and monensin at 37 °C for 4 hours followed by surface and intracellular staining with Foxp3/transcription factor staining buffer set (eBioscience) per the manufacturer's protocol. For tetramer staining, cells were first incubated with tetramer for 30 min prior to the addition of antibodies for surface staining. All data were acquired through LSR Fortessa (BD) and analyzed with FlowJoTM or FACS DIVA (BD Biosciences) software.

For flow cytometry analysis, the following antibodies were used: H-2Db (ThermoFisher Scientific: 28-14-8), H-2Kb (BD Biosciences: AF6-88.5), MHC-IA/IE (BD Biosciences: M5/114.15.2), CD11c (ThermoFisher Scientific: N418), CD80 (BD Biosciences: 16-10A1), CD86 (Biolegend: GL-1), XCR1 (Biolegend: ZET), SIRP1*α/CD172a (BD Biosciences: P84)*. H-2kb-SIINFEKL (BioLegend: 25-D1.16), CD90.2 (BD Biosciences: 53-2.1), CD8 (BD Biosciences: 53-6.7), CD4 (BD Biosciences: RM4.5), CD3 (BD

Biosciences: 17A2), CD45 (BD Biosciences: 30-F11), CD45R (BD Biosciences: RA3-6B2), F4/80 (BD Biosciences: T45-2342), CD11b (ThermoFisher Scientific: M1/70), IFNγ (BD Biosciences: XMG1.2), granzyme B (BD Biosciences: GB11), CD44 (BD Biosciences: IM7), CD62L (BD Biosciences: MEL-14), KLRG1 (BD Biosciences: 2F1). CD49a (BD Biosciences: Hα31/8), TIM-1 (BD Biosciences: 5D12), PD-1/CD279 (BD Biosciences: J43), Ki67 (BD Biosciences: B56), CD69 (BD Biosciences: H1.2F3), and IL-12p40/p70 (BD Biosciences: C15.6). For tetramer staining, iTAg Tetramer/PE against H-2 Kb OVA SIINFEKL (MBL TB-5001-1) and H-2 Kb-restricted MuLV p15E KSPWFTTL (MBL TS-M507-1) were used.

## Immunoblotting
Whole cell lysates were prepared in RIPA lysis buffer (ThermoFisher Scientific: 89900) with Halt™ Protease Inhibitor Cocktail (ThermoFisher Scientific: 78429). Protein concentrations were quantified with the Pierce BSA Standard Pre-Diluted Set (ThermoFisher: 23208). Samples were denatured in NuPage 1× LDS/reducing agent buffer for 10 min at 95 °C. 30–60 µg protein samples were loaded into 4–12% Bis-Tris gels and transferred onto nitrocellulose membrane (ThermoFisher Scientific: 88018) using the BioRad Trans-blot Turbo System. Membranes were blocked with 5% non-fat dry milk and incubated with primary antibodies overnight at 4 °C and then incubated with host species-matched HRP-conjugated secondary antibodies (BioRad: STAR207P, 5184-2504, STAR208P at 1:10,000 dilution) for 1 h at room temperature. Membranes were developed using chemiluminescence (Pierce™ ECL Western Blotting Substrates, SuperSignal™ West Femto Maximum Sensitivity Substrate, Amersham ECL Prime) and detected using Li-Cor.

For immunoblot analysis, the following primary antibodies were used at 1:1000 dilution unless otherwise specified: MHC-I (ThermoFisher Scientific: PA5-115363 at 1:100 dilution), MHC-IA/IE (ThermoFisher Scientific: M5/114.15.2 at 1:100 dilution), CD86 (Cell Signaling: E5W6H), CD80 (Cell Signaling: E6J6N), CD40 (Cell Signaling: E2Z7), p-NF-κB p65 Ser536 (Cell Signaling: 93H1), NF-κB p65 (Cell Signaling: D14E12), p- IκB-α Ser32/36 (Cell Signaling: 5A5), IκB-α (Cell Signaling: 9242), IκB-β (Cell Signaling: D1T3Z), IKK-α (Cell Signaling: D3W6N), IKK-β (Cell Signaling: 8943), IKK-γ (Cell Signaling: 2685), IKK-ε (Cell Signaling: D61F9), p-TBK-1 (Cell Signaling: D52C2), TBK-1 (Cell Signaling: D1B4), vinculin (Sigma Aldrich: V9131 at 1:2000 dilution), and total histone 3 (Cell Signaling: 96C10).

## Preparation and analysis of bulk RNA-sequencing data
Total RNA was extracted from cDCs using the miRNeasy mini kit with the inclusion of the genomic DNA digestion step with the RNase-free DNase Kit (Qiagen). RNA quality was assessed by the Bioanalyzer RNA Nano Chip and depletion of rRNA prior to library generation was performed using RiboErase selection kit (Cat.# KK8561, Kapa Biosystems). Then, the KAPA RNA HyperPrep Kit (Cat.# KK8541, Roche Sequencing Solutions) was used to generate libraries, and sequencing was performed on the Illumina HiSeq™ 2500. Reads were aligned with the Spliced Transcripts Alignment to a Reference (STAR) to the mouse reference genome mm1068. Tables of read counts were then transformed into fragments per kilobase of transcript per million mapped reads using the Bioconductor edgeR[106] package (3.32.1) in R for further downstream analysis. Differential expression analyses were performed using the Bioconductor limma[110] package (3.36.0) in R. Gene Set Enrichment Analysis was performed using the fgsea[111] package (1.16.0).

## Analysis of published bulk RNA-sequencing data
The Riaz et al. dataset[44] was downloaded as FPKM expression from Gene Expression Omnibus (GEO) (GSE91061) and github. The TCGA Pan-Cancer dataset was downloaded as FPKM expression from the publications summary website [https://gdc.cancer.gov/about-data/publications/pancanatlas]. The Murakami et al. dataset[81] was downloaded as TPM expression from GEO (GSE149761)

## Analysis of published single cell RNA-sequencing data
We identified human cDCs in all scRNA-seq datasets as previously validated[16] (*C1orf54, CPVL, LGALS2, CA2, PAK1, CLEC10A, HLA-DMA, HLA-DQB1, HLA-DRA, HLA-DRB1, LYZ, FSCN1*). Single-cell analysis was performed using the Seurat package (4.1.1)[112].

The Qian et al. dataset[48] (breast, colorectal, lung, and ovarian cancers) was downloaded as raw counts from the authors' website [https://lambrechtslab.sites.vib.be/en]. Log1p data normalization and clustering was performed using the unsupervised graph-based clustering approach. The *AverageExpression* function was employed to extract the cell type-specific cluster gene expression.

The Sade-Feldman et al. melanoma dataset[49] was downloaded as log2(TPM + 1) expression from the Human Cell Atlas website [https://www.humancellatlas.org/]. The data was subsetted for "pre-treated" samples only. The *FetchData* function was utilized to evaluate gene expression per cell type.

The Chow et al. endometrial cancer dataset[50] was downloaded as raw counts from GEO (GSE212217) Log1p data normalization and clustering was performed using the unsupervised graph-based clustering approach. The data was subsetted for "pre-treated" samples only.

## Animal experiments
All animal studies were conducted with the approval of the Institutional Animal Care and Use Committee at the University of Michigan prior to initiation of procedures and data collection. Female or male wild type C57BL/6J mice (Stock #: 000664), *Pikfyve*^f/f^ mice (Stock #: 029331), *Itgax*^Tg/0^ (Stock #: 007567) mice, OT-I TCR transgenic mice (Stock #:003831), OT-II TCR transgenic mice (Stock #:004194), *Batf3*^−/−^ mice (Stock #: 013755), and NSG™ (Stock #: 005557) were purchased from The Jackson Laboratory. *Itgax*^Tg/0^ *Pikfyve*^f/f^ C57BL/6 mice were bred internally and genotyped according to the standard protocol provided by The Jackson Laboratory. All mice were housed under specific pathogen-free conditions. All studies were compliant with all relevant ethical regulations regarding animal research. Tumor volumes were measured by caliper every 2–3 days, and tumors did not exceed 2 centimeters in any direction, have ulceration greater than half the surface area of the tumor or effusions, hemorrhage or infections, or allowed to limit physiologic function of the mice in accordance with the committee's Tumor Burden Policy for Rodents. All mice were euthanized with $CO_2$ inhalation as the primary method and bilateral pneumothorax as the secondary method. All mice were maintained under specific pathogen-free conditions and co-housed under standard dark/light cycles, humidity and temperature conditions which were not change throughout the study.

For in vivo studies in genetic conditional knock-out models, 8- to 12-week-old male and female sex-matched littermate pairs of *Pikfyve*^f/f^ and *Itgax*^Tg/0^ *Pikfyve*^f/f^ were used for all tumor studies. For the MC38 and MC38-OVA tumor models, $1.5 \times 10^6$ tumor cells were subcutaneously injected into both flanks of male mice. For the B16F10 and B16F10-OVA tumor models, $0.5 \times 10^6$ tumor cells were subcutaneously injected into both flanks of female mice. For the MCA-205 tumor model, $1 \times 10^6$ tumor cells were subcutaneously injected into both flanks of male mice. For the anti-PD-1 therapy study, on day 5 following inoculation with B16F10 tumor, 200 µg isotype control antibody (Bio X Cell: BE0089, clone 2A3) or 200 µg anti-PD-1 (Bio X Cell: BE0146, clone RMP1-14) was administered intraperitoneally to each mouse every 3 days throughout the experiment. For in vivo assessment of IL-12 expression in DCs, on day 11 following inoculation with MC38 tumor, immune cells were isolated from tumor-draining lymph nodes with Ficoll and incubated for 20 h with 20 ng/ml LPS in 10% FBS medium prior to 5 h stimulation for cytokine staining.

For drug therapy studies, 6- to 8-week-old male and female wild type C57BL/6 mice were inoculated with MC38 or B16F10 syngeneic cancer cell lines. For the anti-IFNγ blockade study, on day 7 following inoculation with MC38 tumor, 100 μg isotype control antibody (Bio X Cell: BE0088, clone HRPN) or 100 μg mouse anti-IFNγ (Bio X Cell: BE0055, Clone XMG1.2) was administered intraperitoneally in 4 doses every other day to each mouse with either vehicle or apilimod 30 mg/kg given either 5 days per week (MC38) or daily (B16F10) administered by oral gavage. For the anti-IL-12 blockade study, on day 7 following inoculation with MC38 tumor, a single dose of 1 mg isotype control antibody (Bio X Cell: BE0089, clone 2A3) or 1 mg mouse anti-IL-12p40 (Bio X Cell: BE0051, Clone C17.8) was administered intraperitoneally to each mouse followed by three additional doses of 500 μg of either antibody given every other day± either vehicle or apilimod (30 mg/kg, 5 days per week) administered by oral gavage.

For vaccine strategy experiments, 6- to 8-week-old female wild type C57BL/6 mice were inoculated with B16F10-OVA tumors. For the vaccination study, mice were pretreated with vehicle or apilimod (30 mg/kg, 5 days per week) administered by oral gavage ± water or PolyI:C subcutaneously (100 μg on Day 1 and Day 14) for 21 days. On Day 22, B16F10-OVA tumors were inoculated and monitored for an additional 14 days in the absence of treatment. For the vaccine therapy study, B16F10-OVA tumors were inoculated. On Day 5, mice began treatment with vehicle or apilimod (30 mg/kg, 5 days per week) administered by oral gavage ± water or PolyI:C subcutaneously (100 μg on Day 5 and Day 12) for a total of eleven days of treatment.

### Statistics and reproducibility

No statistical method was used to predetermine sample sizes. Human sequencing analyses were limited to pre-treatment tumor samples. Otherwise, no other data were excluded from the analyses. For in vitro studies, treatment groups were randomly assigned at the time freshly isolated bone marrow cells were plated and were not changed when treatment was given on the culture day indicated. These experiments were completed in replicates and independent experiments. For animal studies, mice were randomly assigned to treatment groups after tumor inoculation. The starting tumor burden in the treatment and control groups was similar before treatment. The investigators were not blinded to allocation during experiments and outcome assessment.

Overall survival was estimated by Kaplan–Meier methods and compared with log-rank tests. Cox proportional hazard models were used for multivariate survival analysis. Multivariate logistic regression models were used to assess binary outcomes of response to treatment. Pearson's correlation coefficient was used to assess linear correlations between variables. Chi-square test was performed to assess independence of groups. All statistical analyses were performed using R packages tidyverse (1.3.1), dplyr (1.0.7), ggplot2 (3.3.5), survminer (0.4.9) and survival (0.4.9).

In vitro experiments were carried out in duplicate or triplicate independent experiments as indicated. Student $t$-tests were performed to assess differences between groups. All in vivo studies included a minimum of three littermate pairs of WT or KO mice as biological replicates for genetic conditional knock-out studies and a minimum of four mice per group for drug therapy studies. Mann–Whitney $U$ or ANOVA with post-hoc Tukey adjustment for multiple comparisons were performed to assess differences between groups as indicated. All statistical analyses were performed using GraphPad Prism 9 software.

### Reporting summary

Further information on research design is available in the Nature Portfolio Reporting Summary linked to this article.

## Data availability

The source data generated in this study are included in the Source Data, Supplementary Information, or Supplementary Data; Supplementary Data 9 contains all source data for the Supplementary Information. All materials are available from the corresponding authors upon reasonable request. Clinical sequencing data are publicly available with raw data available upon request from dbGaP phs000673.v5.p1 [https://www.ncbi.nlm.nih.gov/projects/gap/cgi-bin/study.cgi?study_id=phs000673.v5.p1][40,43]. RNA-seq data newly generated in this study for in vitro analysis have been deposited in the GEO repository at NCBI under accession codes GSE235596 and GSE235599. The accession code fo the Riaz et al. dataset[44] is GSE91061. and also available on github [https://github.com/riazn/bms038_analysis]. The TCGA Pan-Cancer dataset is available on the publications summary website [https://gdc.cancer.gov/about-data/publications/pancanatlas]. The accession code for the Murakami et al. dataset[81] is GSE149761. The Qian et al. dataset[48] is available on the authors' website [https://lambrechtslab.sites.vib.be/en]. The Sade-Feldman et al. melanoma dataset[49] is available on the Human Cell Atlas website [https://www.humancellatlas.org/]. The accession code for the Chow et al. endometrial cancer dataset[50] is GSE212217. Source data are provided with this paper.

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

## Acknowledgements

We thank members of the Chinnaiyan and Zou/Kryczek labs in the Michigan Center for Translational Pathology for their support and intellectual contributions to this study. We thank Stephanie Miner for assistance with the review, editing, and submission of this paper. We also thank Stephanie Simko, Andrew Delekta, Ingrid Apel, and Mishaal Yazdani for their technical assistance and Wojtek Szeliga,Teri Elkins, Chia-Mei Huang, and Melissa Dunn for their assistance with lab management. Finally, we thank Goutham Narla and James Moon for their scholarly contributions and discussions. This work was supported by the NCI Oustanding Investigator Award (R35 CA231996) and the NCI Prostate SPORE (P50 CA186786) grants to A.M.C. and NCI F31 (CA26461-01) grant to J.E.C. A.M.C. is a Howard Hughes Medical Institute Investigator, A. Alfred Taubman Scholar, and American Cancer Society Professor.

## Author contributions

J.E.C., W.Z., and A.M.C. conceived the idea, designed the experiments, and composed the paper; J.E.C. conducted all experiments with assistance from J.Y., J.G., T.M., and S.Y.; Y.Q. and Y. Zheng assisted with vaccine experiments; I.K., J.Y., Y.B., and T.M. assisted with flow cytometry experiments and analysis; I.K., J.Y., Y.B., T.M., L.V., H.L., and M.D.G. assisted with immunological experimental techniques; I.K., J.Y., T.M., A.P., H.X., J.Z., H.L., G.L., and M.D.G. assisted with interpretation of results; Y.Q., J.G., Y.B., J.C.T., S.W., S.G., L.V., Y. Zheng, and T.H. assisted with animal experiments. M.G., Y. Zhang, X.C., F.S., R.W., and M.C. assisted with RNA-seq datasets analyses; M.C. and M.D.G. assisted with clinical analysis and the interpretation of the results. A.M.C. supervised the project.

## Competing interests

A.M.C. is co-founder and SAB member of LynxDx, Esanik, Medsyn, and Flamingo Therapeutics. A.M.C. serves as a scientific advisor or consultant to EdenRoc, Proteovant, Rebus, and Tempus. W.Z. has served as a scientific advisor or consultant for NGM, CrownBio, Cstone, Proteo-Vant, Hengenix, NextCure, and Intergalactic. J.E.C., A.M.C., and W.Z. have submitted a patent application with the University of Michigan, Ann Arbor. The remaining authors declare no competing interests.
