## [Peer Review File · Nature Communications]

PIKfyve, expressed by CD11c-positive cells, controls tumor immunityEditorial Note: This manuscript has been previously reviewed at another. This document only contains reviewer comments and rebuttal letters for versions considered at *Nature Communications*.

REVIEWERS' COMMENTS

Reviewer #1 (Remarks to the Author):

The authors have provided compelling new data that addresses the questions raised in the last review. I have no further concerns and thank the authors for taking the time to address these aspects.

Reviewer #2 (Remarks to the Author):

The authors made significant efforts to address the comments of the reviewer on the original manuscript.

The authors show data suggesting that PYKfive affects DC activation in the tumor context (new data in Fig. 4) and how the genetic and pharmacological inhibition increases immunity against cancer. They also explored the potential effect in T cell priming and a potential link to NFkB activation.

The minor remaining issue in supp Fig. 1f is addressed.

REVIEWERS' COMMENTS

Reviewer #1 (Remarks to the Author):

The authors have provided compelling new data that addresses the questions raised in the last review. I have no further concerns and thank the authors for taking the time to address these aspects.

Reviewer #2 (Remarks to the Author):

The authors made significant efforts to address the comments of the reviewer on the original manuscript.

The authors show data suggesting that PYKfive affects DC activation in the tumor context (new data in Fig. 4) and how the genetic and pharmacological inhibition increases immunity against cancer. They also explored the potential effect in T cell priming and a potential link to NFkB activation.

The minor remaining issue in supp Fig. 1f is addressed.

AUTHOR RESPONSE

We thank the reviewers for their thoughtful and instructive comments that have significantly improved this manuscript. We appreciate their acknowledgement of our efforts to enhance this work and provide a meaningful contribution to the Nature Communications readership.